

# Effects of dietary supplementation with *Lactobacillus acidophilus* on the performance, intestinal physical barrier function, and the expression of NOD-like receptors in weaned piglets

Shiqiong Wang[1], Haihua Li[2], Chenhong Du[1], Qian Liu[1], Dongji Yang[1], Longbin Chen[3], Qi Zhu[3] and Zhixiang Wang[1]

[1] College of Animal Science and Veterinary Medicine, Henan Agricultural University, Zhengzhou, China
[2] College of Animal Science and Veterinary Medicine, Tianjin Agricultural University, Tianjin, China
[3] Tianjin Institute of Animal Husbandry and Veterinary Medicine, Tianjin Academy of Agricultural Sciences, Tianjin, China

## ABSTRACT

*Lactobacillus* supplementation is beneficial to the barrier function of the intestinal physical barrier in piglets. However, the mechanisms underlying this beneficial function remain largely unknown. Here, we investigated the effects of dietary supplementation with *Lactobacillus acidophilus* on the performance, intestinal physical barrier functioning, and NOD-like receptors (NLRs) expression in weaned piglets. Sixteen weaned piglets were randomly allocated to two groups. The control group received a corn-soybean basal diet, while the treatment group received the same diet adding 0.1% *L. acidophilus*, for 14 days. As a result, dietary *L. acidophilus* supplementation was found to increase the average daily gain (ADG) ($P < 0.05$), reduced serum diamine oxidase (DAO) activity ($P < 0.05$), increased the mRNA expression and protein abundance of occludin in the jejunum and ileum ($P < 0.01$), reduced the mRNA levels of NOD1 ($P < 0.01$), receptor interacting serine/threonine kinase 2 (RIPK2) ($P < 0.05$), nuclear factor κB (NF-κB) ($P < 0.01$), NLR family pyrin domain containing 3 (NLRP3) ($P < 0.01$), caspase-1 ($P < 0.01$), interleukin 1β (IL-1β) ($P < 0.05$) and IL-18 ($P < 0.01$) in the jejunum tissues of the weaned pigs. The expression of NLRP3 ($P < 0.05$), caspase-1 ($P < 0.01$), IL-1β ($P < 0.05$) and IL-18 ($P < 0.05$) was also reduced in the ileum tissues of the weaned pigs. These results showed that *L. acidophilus* supplementation improves the growth performance, enhances the intestinal physical barrier function, and inhibits the expression of NOD1 and NLRP3 signaling-pathway-related genes in jejunum and ileum tissues. They also suggest that *L. acidophilus* enhances the intestinal physical barrier functioning by inhibiting IL-1β and IL-18 pro-inflammatory cytokines via the NOD1/NLRP3 signaling pathway in weaned piglets.

Corresponding authors
Haihua Li, lihaihuaok@126.com
Zhixiang Wang, wzxhau@aliyun.com

## INTRODUCTION

As the interface between the external environment and the internal milieu, the physical barrier function of the intestine is extremely important (*Sperandio, Fischer & Sansonetti, 2015*). The integrity of the physical epithelial barrier function depends on the presence of healthy epithelial cells and a normally functioning paracellular pathway. The paracellular pathway is a complex structure predominantly controlled by tight junctions (TJs) between the epithelial cells, which form selective channels that regulate the selective permeability of the intestinal mucosa to endotoxins, bacteria, water, ions, and nutrients (*Balda & Matter, 2016*); therefore, the gut has barrier function. Impaired intestinal physical barrier leads to bacterial translocation and increased intestinal permeability (*Wang et al., 2014*). In mammals, D-lactate originates from the metabolic reactions of intestinal microorganisms because the rapid metabolic enzyme systems that produce D-lactate are absent in them. Therefore, increased serum D-lactic acid concentrations may reflect a dysfunctional intestinal mucosal barrier (*Vella & Farrugia, 1998*). DAO, a highly active endoenzyme in the microvilli of the mammalian small intestine, regulates cell proliferation by degrading polyamine, a substance essential for mitosis and meiosis (*Peng et al., 2004*), and its activity in the blood can be exploited as a useful marker for the integrity of or damage to the intestinal mucosa (*Li et al., 2002*).

TJs mainly comprise transmembrane proteins such as occludin, claudins, junctional adhesion molecules, zonula occludens (ZO), and other peripheral cytoplasmic proteins (*Balda & Matter, 2016*). The occludin protein blocks the gap between cells, forms a TJ structure with the ZO-1 protein, and is expressed in virtually all epithelial and endothelial tissues. Its functions include the maintenance of cell polarity, and the regulation of signal molecule localization and cell permeability (*Mariano et al., 2011*). TJs act as signaling hubs that direct epithelial proliferation and differentiation (*Xie et al., 2016*; *Volksdorf et al., 2017*), the study found that occludin is involved in immune signaling pathways, making it important in cellular physiology (*Luo et al., 2017*). ZO provides the basis of the structural support for TJs (*Rodgers et al., 2013*). In addition, the structure and function of TJs depends mainly on the actin cytoskeleton, and these junctions anchor the actin cytoskeleton in the epithelial cells through ZO, with peripheral actin present in the cell connection ring around the top of the cell (*Balda & Matter, 2016*). This structure not only provides a barrier function for the epithelial cells, but also changes the membrane permeability via its rapid assembly and disassembly in response to different stimuli (*Koch & Nusrat, 2009*). Thus ZO, which is considered to be a major moderator of TJ permeability (*Li et al., 2016*), can reversibly modulate intestinal permeability.

Previous studies have shown that disrupting the TJ structure in the mucosal barrier damages the intestinal epithelial monolayer and increases its permeability (*Tanaka et al., 2015*), and intestinal permeability is exquisitely sensitive to immune cell signaling. As microbial sensors, the NOD-like receptors (NLRs) are thought to recognize and resist microbial pathogens and act to regulate the intestinal flora and intestinal homeostasis. The NLR family contains more than 20 cytosolic receptors in mammals (*Wells et al., 2011*), and these proteins are involved in complex signaling pathways. NLRs activate the

host's defense mechanisms through two major signaling pathways. One detects bacterial peptidoglycan through NOD1 or NOD2 signals, activates NF-κB and MAPK via the serine/threonine kinase RIP2 (RICK/CARDIAK/RIPK2), and causes the release of pro-inflammatory cytokines and chemokines, such as tumor necrosis factor α (TNF-α), IL-12 and IL-8 (*Kobayashi et al., 2002*). In the other pathway, caspase-1 is activated through several NLR signals, which leads to IL-1β secretion and programmed cell death. Recent studies have shown that NLR-mediated signaling pathways are involved in regulating epithelial barrier repair and integrity (*Lei-Leston, Murphy & Maloy, 2017*).

A number of research studies have reported that *Lactobacillus* can stimulate immune cells in a specific way that enhances their response functions, as well as acting to maintain a normal immune response, regulate cytokine release (*Liu et al., 2011*), reduce harmful or excessive inflammation (*Ménard et al., 2004*), and maintain the intestinal barrier function (*Laval et al., 2015*). Does *Lactobacillus* maintain the intestinal barrier function via NLR-mediated signaling pathways? Few studies have investigated this question in weaned piglets. Therefore, we evaluated the effects of a newly isolated *L. acidophilus* strain on the growth performance, intestinal barrier function, and the expression of NLR signaling pathway-related genes in weaned piglets. Our results indicate that dietary supplementation with *L. acidophilus* enhances intestinal barrier functioning by inhibiting IL-1 β and IL-18 pro-inflammatory cytokines via the NOD1/NLRP3 signaling pathway in the weaned piglets.

## MATERIALS AND METHODS

The animal care and use protocols were approved by the Review Committee for the Use of Institutional Animal Care and Use Committee of Henan Agricultural University (Protocol no. 20161030).

### *Lactobacillus acidophilus* strain

The *L. acidophilus* strain used in this study was isolated from the excrement of a healthy piglet. The bacterium was cultured in MRS medium under anaerobic conditions at 37 °C for 24 h, and the culture medium was centrifuged at $3,000 \times$ g for 10 min at 4 °C. The strain was powdered by vacuum freeze drying (Tofflon, Shanghai, China) at a concentration of $5 \times 10^{10}$ colony-forming units (CFU)/g of freeze-dried *Lactobacillus* powder. The bacterial concentration was measured with an ultraviolet (UV) spectrophotometer (Nano Drop; Thermo Fisher, Waltham, MA, USA) at a wavelength of 550 nm.

### Animals and experimental design

Sixteen crossbred (Duroc × Large White × Landrace) healthy piglets were weaned at 28 days of age. After acclimatization for 3 days, the piglets were stochastically divided on the basis of their initial bodyweights into two groups into eight replicate pens ($n = 8$ each). (1) In the control group, the piglets were fed a corn-soybean basal diet. (2) In the treatment group, the piglets were fed the same diet supplemented with 0.1% *L. acidophilus* (*Qiao et al., 2015*). The animals were housed individually in metabolic cages ($1.0 \times 0.5 \times 0.8$ m$^3$) with natural light, and fed four times daily (08:00, 12:00, 16:00, and 20:00) for 14 days,

**Table 1  Composition and nutrient contents of the basal diets (%w/w, as-fed basis).**

| Item | Amount |
|---|---|
| Ingredients, % | |
| Corn | 63.20 |
| Soybean meal, 43% CP (crude protein) | 19.00 |
| Whey powder | 4.80 |
| Fish meal, 65% CP | 8.60 |
| Glucose | 1.00 |
| Acidifier | 0.30 |
| Calcium hydrogen phosphate | 0.60 |
| Limestone | 0.70 |
| Salt | 0.30 |
| L-Lys HCL, 78% Lys | 0.30 |
| DL-Met, 99% Met | 0.10 |
| L-Thr, 98% Thr | 0.10 |
| Vitamin and mineral premix[a] | 1.00 |
| Calculated composition | |
| DE (digestible energy), Mcal/kg | 3.25 |
| Lys (%) | 1.39 |
| Met (%) | 0.53 |
| Analyzed composition | |
| Crude protein | 18.65 |
| Calcium | 0.85 |
| Total phosphorus | 0.69 |

**Notes.**

[a]Provided the following per kg of diet: Vitamin A, 12,500 IU; Vitamin D, 1250 IU; Vitamin E, 125 IU; Vitamin $B_{12}$, 90 $\mu$g; Vitamin $B_2$, 10 mg; Pantothenic acid, 48 mg; Niacin, 35 mg; Folic acid, 4.5 mg; Biotin, 0.25 mg; Fe, 130 mg; Zn, 180 mg; Cu, 15 mg; Mn, 30 mg; I, 0.60 mg; Se, 0.25 mg.

with water provided *ad libitum*. The room temperature was maintained at 25–28 °C. The basic diet composition and nutrient levels are shown in Table 1.

## Sample collection

At the end of the trial, all piglets were weighed. The final body weight (FBW), average daily feed intake (ADFI), average daily gain (ADG), and feed/gain ratio (F/G) were recorded during the trial period. The blood samples were collected from them through the precaval. Then, three piglets randomly selected from each group were killed by an intracardiac injection of sodium pentobarbital (50 mg/kg bodyweight). The abdominal cavity was opened immediately, and the jejunum and ileum tissues were collected, washed with phosphate-buffered saline, immediately frozen in liquid nitrogen, and stored at −80 °C for downstream analyses. The jejunum and ileum were collected at the same site in all the animals. Sera were obtained by centrifuging the blood samples at $3,000 \times$ g for 20 min at 4 °C followed by storage at −20 °C until use.

**Table 2 Amplification reaction system of qPCR.**

| Component | Volume ($\mu$L) |
|---|---|
| 2 × All-in-One qPCR Mix | 10 $\mu$L |
| PCR forward primer (2 $\mu$M) | 2 $\mu$L |
| PCR reverse primer (2 $\mu$M) | 2 $\mu$L |
| Template[c] | 2 $\mu$L |
| RNase Free Water | Up to 20 $\mu$L |

**Table 3 Amplification program of qPCR.**

| | Cycle | Temperature | Time |
|---|---|---|---|
| Preincubation | 1 | 95 °C | 10 min |
| | | 95 °C | 10 s |
| Amplification | 40 | 60 °C | 20 s |
| | | 72 °C | 15 s |

## Serum diamine oxidase (DAO) and D-lactic acid measurements

Serum DAO was analyzed using a commercial kit (number A088-1; Jiancheng Bioengineering Institute, Nanjing, China) and serum D-lactic acid was analyzed using a commercial enzyme-linked immunosorbent assay kit (number H263; Jiancheng Bioengineering Institute, Nanjing, China).

## Real-time quantitative PCR analyses

Total RNA was extracted from the jejunum and ileum tissues using TRIzol reagent (Invitrogen, USA), according to the manufacturer's instructions. A UV spectrophotometer (UV754N, Shenzhen, China) was used to determine the RNA concentrations based on their absorbances at 260 nm ($A_{260}$). We assessed the purity of the RNA by measuring the $A_{260}/A_{280}$ ratio, and the RNA samples typically had $OD_{260}/OD_{280}$ ratios of 1.9–2.0. The integrity of the RNA was analyzed by electrophoresis on a 1% agarose gel. The PCR templates were reverse transcribed using the All-in-One$^{TM}$ First-Strand cDNA Synthesis Kit (GeneCopoeia, USA), according to the manufacturer's instructions. Following reverse transcription, real-time quantitative PCR with SYBR All-in-One$^{TM}$ qPCR Mix (GeneCopoeia) and the LightCycler-$^{®}$ 96 Real-Time PCR System (Roche, Switzerland) was used to quantify the expression levels of the occludin, ZO-1, NOD1, receptor interacting serine/threonine kinase 2 (RIPK2), NLR family pyrin domain containing 3 (NLRP3), caspase-1, nuclear factor (NF)-$\kappa$B, interleukin (IL)-1$\beta$, IL-18 and $\beta$-actin mRNAs in the jejunum and ileum tissues. The reaction system and program are shown in Tables 2 and 3.

The primers were synthesized by the Shanghai Biological Technology Co. (Shanghai, China), and their characteristics are listed in Table 4. The occludin, ZO-1, NOD1, RIPK2, NLRP3, caspase-1, IL-1$\beta$, and IL-18 primers were designed according to their respective cDNA sequences in GenBank, namely NM_001163647.2, XM_013993251.1, NM_001114277.1, XM_021089139.1, NM_001256770.2, NM_214162.1, NM_214055.1, and NM_213997.1. The primers of NF-$\kappa$B and $\beta$-actin cDNA were used as previously

**Table 4  Primer sequences used for the real-time quantitative PCR.**

| Gene | Primer sequences 5′–3′ | Product size (bp) | $T_m$ (°C) |
|------|------------------------|-------------------|------------|
| *Occludin* | F: CAGGTGCACCCTCCAGATTG<br>R: ATGTCGTTGCTGGGTGCATA | 167 | 60 |
| *ZO-1* | F: TGCTGGTTTGAAGCCTCCAG<br>R: GGGGTTTTTGAGGTTCTGGC | 134 | 60 |
| *NOD1* | F: ACTGACAGTGGGGTGAAGGT<br>R: TTTCCCAGTTTCAGGCACTTG | 158 | 60 |
| *RIPK2* | F: GTGGATGGGCACAAAATCCAG<br>R: TGGAAGCACTTTGCAACTTTGT | 144 | 60 |
| *NLRP3* | F: TTTGGCTGTTCCTGAGGCAG<br>R: AGGGCATAGGTCCACACAAA | 105 | 60 |
| *caspase-1* | F: CGAACTCTCCACAGGTTCACAA<br>R: AAGCTTGAGGCTCCCTCTTG | 145 | 60 |
| *IL-1β* | F: GAAAGCCCAATTCAGGGACC<br>R: TGCAGCACTTCATCTCTTTGG | 172 | 60 |
| *IL-18* | F: GTAGCTGAAAACGATGAAGACCTG<br>R: GGCATATCCTCAAACACGGC | 134 | 60 |
| *NF-κB* | F: AGTACCCTGAGGCTATAACTCGC<br>R: TCCGCAATGGAGGAGAAGTC | 133 | 60 |
| *β-actin* | F: CTTCCTGGGCATGGAGTCC<br>R: GGCGCGATGATCTTGATCTTC | 201 | 60 |

described (*Chen et al., 2016*; *Luo et al., 2017*). Three replicates were performed for each reaction and the results were expressed according to their $2^{-\Delta\Delta CT}$ values.

## Western blot analysis

The tissue samples (50–100 mg) were homogenized in 1 mL of RIPA Lysis buffer (BOSTER, Wuhan, China) supplemented with protease inhibitors (BOSTER, Wuhan, China) and centrifuged at $12,000 \times$ g for 15 min at 4 °C. Tissue proteins were separated by 10% sodium dodecyl sulfate-polyacrylamide gel electrophoresis (SDS-PAGE) and transferred onto the PVDF membranes (biosharp, China). After blocking with 5% non-fat milk in tris-buffer saline solution buffer (TBST), membranes were incubated with a primary antibodies occludin (Proteintech, Wuhan, China), ZO-1 (AVIVA, Des Moines, IA, USA) or β-actin (Proteintech, Wuhan, China) at 4 °C overnight and then washed three times with TBST for 15 min. Membranes were then incubated with secondary antibodies (Proteintech, Wuhan, China) at room temperature for 1 h. Proteins were visualized using the ECL reagent (BOSTER, Wuhan, China) according to the manufacturer's instructions. Qualification of band intensity was determined with the use of Image J software and normalized to β-actin. Antibodies information are shown in Table 5.

## Statistical analysis

The results are expressed as means ± standard deviations, were analyzed with an independent samples $t$-test, using the SPSS 20.0 software, and statistically significant differences were tested at the 0.05 or 0.01 level.

**Table 5  Antibodies message.**

| Antibodies | Catalog number | Source | Dilution | Company |
|---|---|---|---|---|
| occludin | 13409-1-AP | Rabbit | 1:1,500 | Proteintech, China |
| ZO-1 | ARP36636-P050 | Rabbit | 1:250 | AVIVA, UAS |
| $\beta$-actin | 60008-1-lg | Mouse | 1:10,000 | Proteintech, China |

**Table 6  Effect of dietary treatments on growth performance in weaned piglets.**

| Item | Control | Treatment | $P$ value |
|---|---|---|---|
| IBW (kg) | $7.03 \pm 0.48$ | $7.01 \pm 0.30$ | 0.927 |
| FBW (kg) | $10.83 \pm 0.74$ | $11.16 \pm 0.50$ | 0.322 |
| ADFI (g) | $474.88 \pm 61.00$ | $495.27 \pm 35.63$ | 0.426 |
| ADG (g) | $271.97 \pm 20.27^{a}$ | $296.52 \pm 14.71^{b}$ | 0.015 |
| F/G | $1.74 \pm 0.10$ | $1.67 \pm 0.08$ | 0.139 |

Notes.

    Each value represents the mean SD of eight replicates, values in the same row with no superscript letter are not significantly different ($P > 0.05$), whereas those with different small superscript letters are significantly different ($P < 0.05$).

**Table 7  Effect of dietary treatments on serum DAO activity and D-lactic acid concentration in weaned piglets.**

| Item | Control | Treatment | $P$ value |
|---|---|---|---|
| DAO (U/ml) | $7.40 \pm 0.25^{a}$ | $7.09 \pm 0.22^{b}$ | 0.017 |
| D-lactic acid (mmol/L) | $2.27 \pm 0.12$ | $2.18 \pm 0.10$ | 0.120 |

Notes.

    Each value represents the mean SD of eight replicates, values in the same row with no superscript letter are not significantly different ($P > 0.05$), whereas those with different small superscript letters are significantly different ($P < 0.05$).

# RESULTS

## Growth performance

The growth performance results for the piglets are shown in Table 6. There was no significant difference in IBW ($P = 0.927$) and FBW ($P = 0.322$), the treatment group had a lower average initial weight and higher average final weight (neither significant), which meant that the *L. acidophilus* group had a significantly higher weight gain. *L. acidophilus* supplementation increased the ADG compared with that of the control group ($P = 0.015$). No differences in the ADFI ($P = 0.426$), or F/G ($P = 0.139$) were observed between the two dietary treatment groups.

## Intestinal physical barrier function
### *Serum DAO and D-lactic acid*

The results for serum DAO activity and D-lactic acid concentrations of the piglets are shown in Table 7. Dietary supplementation with *L. acidophilus* was seen to reduce the serum DAO activity ($P = 0.017$) in the piglets compared with those fed on the control diet. No differences in the serum D-lactic acid concentrations ($P = 0.12$) of the pigs were observed between the two dietary treatment groups.
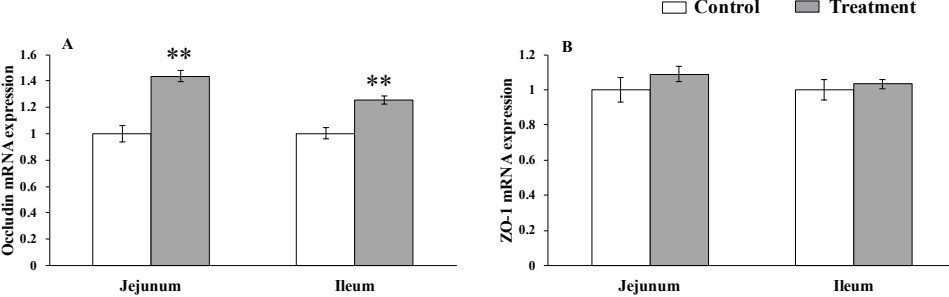

**Figure 1 The relative mRNA expression of TJs in the intestinal tissues of weaned piglets.**

### *Relative mRNA expression levels of TJs*

The results for relative mRNA expression of TJs genes are presented in Fig. 1. *L. acidophilus* supplementation enhanced the mRNA levels of occludin in the jejunum ($P = 0.001$) and ileum ($P = 0.001$) tissues of the weaned pigs compared with those in the control group. No differences were observed in the ZO-1 levels in both of these tissues (jejunum: $P = 0.138$, ileum: $P = 0.43$) from the pigs in the two different dietary treatment groups.

### *Western blot of TJs*

The results for protein expression of occludin and ZO-1 are presented in Fig. 2. The diet supplemented with *L. acidophilus* enhanced the protein levels of occludin in the jejunum ($P < 0.001$) and ileum ($P < 0.001$) tissues of the weaned pigs compared with those in the control group. No differences were observed in the ZO-1 levels in both of these tissues (jejunum: $P = 0.087$, ileum: $P = 0.333$) from the pigs in the two different dietary treatment groups.

## Relative mRNA expression levels of NLR-signaling pathway-related genes

The results for relative mRNA expression of NLR-signaling pathway-related genes are presented in Fig. 3. Dietary *L. acidophilus* supplementation greatly reduced the mRNA levels of NOD1 ($P < 0.001$), NF-kB ($P = 0.001$), NLRP3 ($P = 0.001$), caspase-1 ($P = 0.003$) and IL-18 ($P = 0.004$) and reduced the mRNA levels of RIPK2 ($P = 0.02$) and IL-1 β ($P = 0.011$) in the jejunum tissues of the weaned pigs. Caspase-1 expression was extremely reduced ($P = 0.003$), while that of NLRP3 ($P = 0.025$), IL-1 β ($P = 0.019$) and IL-18 ($P = 0.015$) were reduced in the ileum tissues from the weaned pigs. No differences in NOD1 ($P = 0.603$), RIPK2 ($P = 0.374$), NF-kB ($P = 0.184$) mRNA levels were observed in the ileum tissues from both of the dietary treatment groups.

## DISCUSSION

Some weaned piglets have slow growth, a low feed utilization rate, and gastrointestinal physiological disorders such as diarrhea (*Lallès et al., 2004*). These disorders can be caused by changes in the structure of the gastrointestinal tract or in the intestinal barrier function

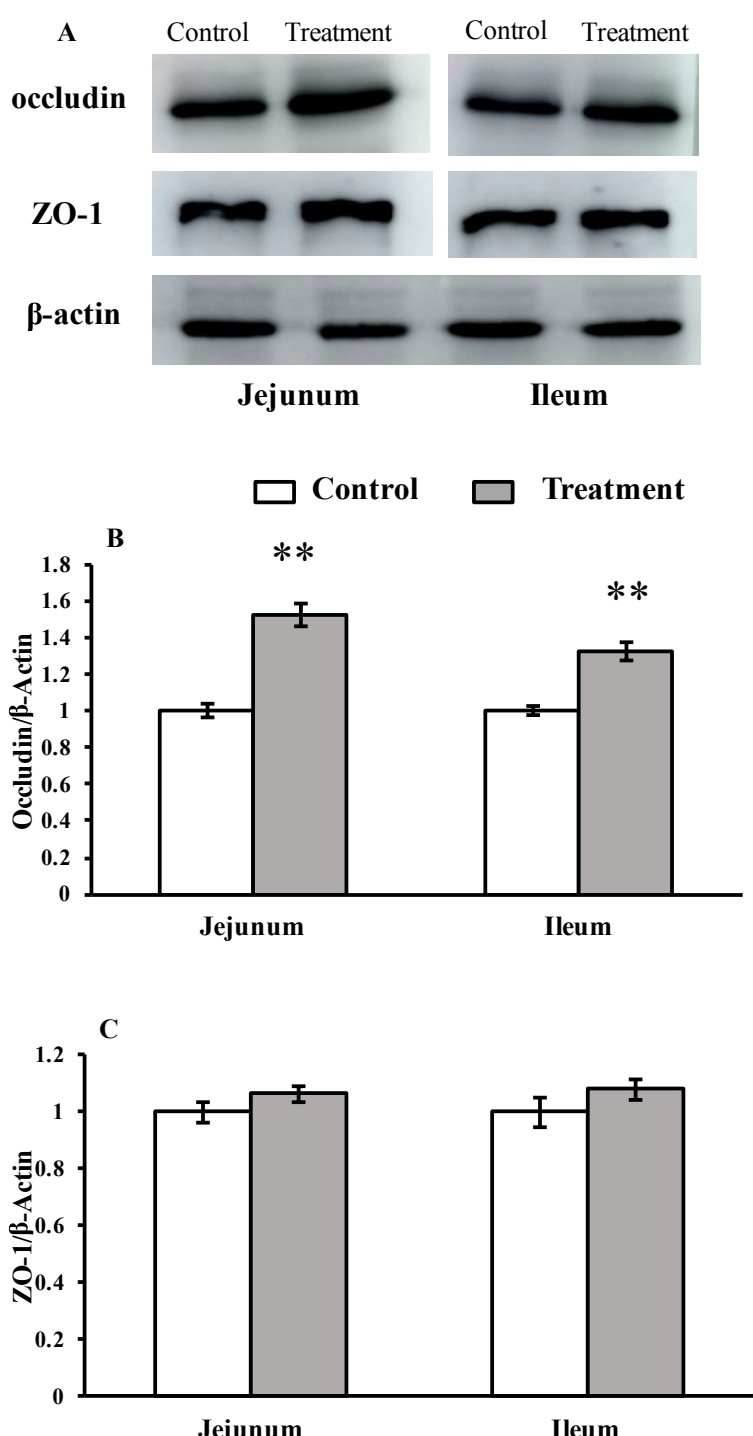

**Figure 2** **The protein abundance of TJs in the intestinal tissues of weaned piglets.** (A) Western blot results for TJs expression. (B) Intensity analysis of occludin was performed and normalized to β-actin. (C) Intensity analysis of ZO-1 was performed and normalized to β-actin.

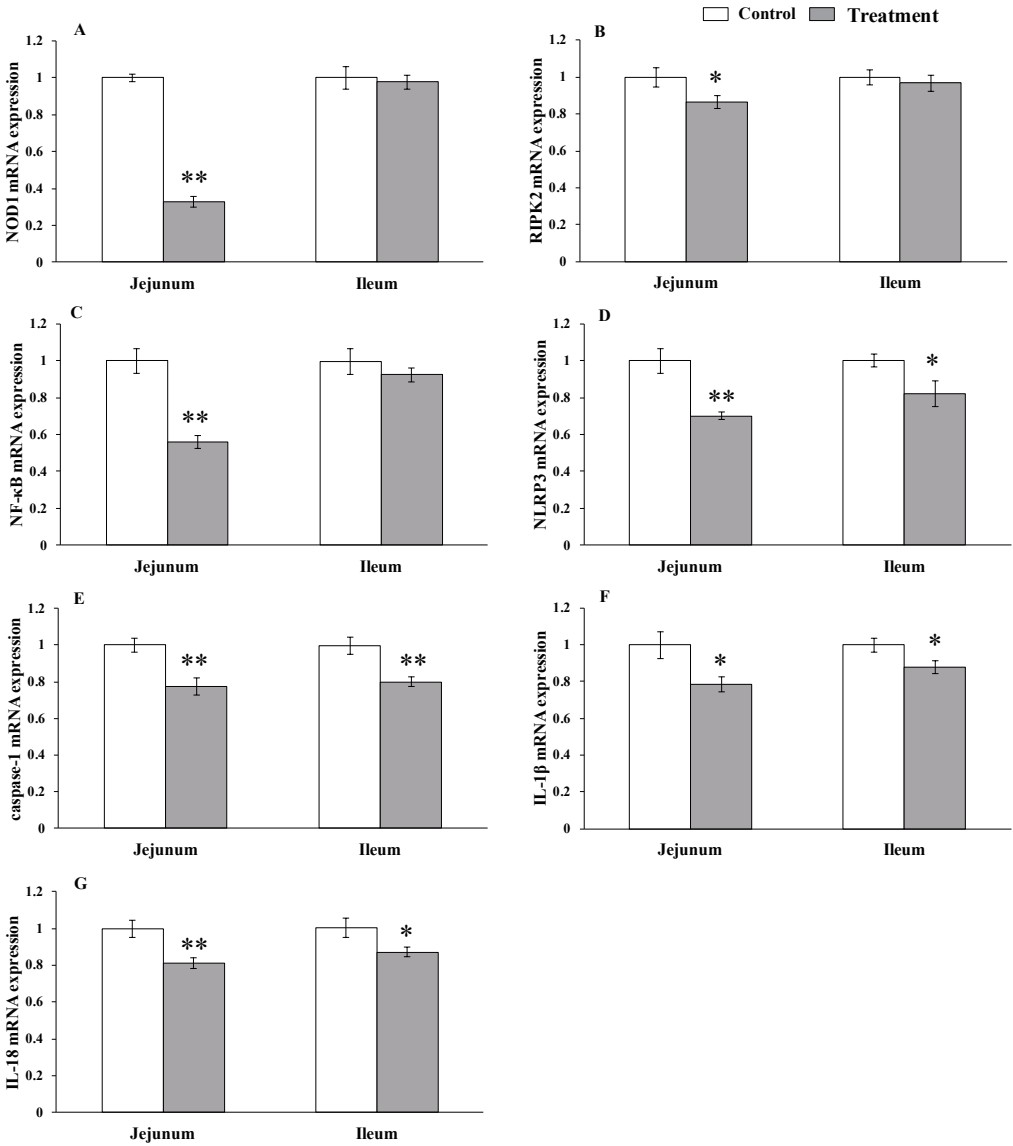

**Figure 3** **The relative mRNA expression of NLR-signaling-pathway-related genes in the intestinal tissues of weaned piglets.** NOD1 (A), RIPK2 (B), NF-$\kappa$B (C), NLRP3 (D), caspase-1 (E), IL-1$\beta$ (F), and IL-18 (G) mRNA relative expression in the jejunum and ileum tissues of weaned piglets. Date represent means ± SD ($n = 3$ per group); bars with asterisk(s) (* or **) indicate a statistically significant difference between control and treatment at $P < 0.05$ or $P < 0.01$ (independent-samples $t$-test).

(*Wijtten, Van der Meulen & Verstegen, 2011*), compromised immune functioning (*Bacou et al., 2017*), or other factors. *Lactobacillus* is the predominant microbe in the intestinal tracts of piglets, where it acts to strengthen the intestinal functions of its hosts (*Wu et al., 2016*) and inhibit the growth of potentially pathogenic bacteria (*Liu et al., 2017a*). The lactic acid produced by the metabolism of lactic acid bacteria can activate pepsin, degrade carbohydrates, and augment the synthesis of amino acids, thereby improving the digestibility of nutrients and promoting piglet growth. *Lan, Koo & Kim (2017)* reported

that supplementation with 1, 2, or 3 g/kg *L. acidophilus* increased ADG in weaning pigs during a 14-day trial. *Wang et al. (2009)* found that the daily gain and feed conversion increased and the occurrence of diarrhea was reduced after the dietary addition of *L. fermentum* I5007 in piglets. These findings are consistent with our results, but other studies have found that *L. acidophilus* supplementation had no effect on ADG, ADFI, or F/G in weanling pigs (*Zhao & Kim, 2015*). These discrepancies may be attributable to differences in the bacterial strains, pig breeds, dose levels, or feeding environments.

Intestinal permeability maintains the organ- and tissue-specific homeostasis required for optimal biological functioning, however, the physiological mechanisms underlying this highly regulated barrier are incompletely understood. *Qiao et al. (2015)* reported that 0.1% or 0.2% *L. acidophilus* supplementation reduced the serum DAO activity in weanling pigs. That our results are consistent with this finding indicates that *L. acidophilus* is beneficial for the integrity of the intestinal mucosa. TJs control the permeability of the gastrointestinal tract (*Tsukita, Furuse & Itoh, 2001*; *Turner, 2009*), and *Lactobacillus* has been shown to enhance the expression of TJs (*Liu et al., 2014*; *Yan & Polk, 2006*). *Yang et al. (2015)* reported that supplementation with *L. reuteri* I5007 dramatically enhanced the abundance of the occludin and ZO-1 proteins in the jejunum and ileum tissues of the piglets in their study. *Mao et al. (2016)* showed that dietary *L. rhamnosus* GG (*LGG*) supplementation increased ZO-1, occludin and Bcl-2 mRNA levels and reduced Bax mRNA levels in the jejunum mucosal tissue from weaned pigs, thus indicating that *LGG* supplementation can improve intestinal permeability by reducing apoptosis. *Roselli et al. (2007)* reported that *L. sobrius* supplementation alleviated the membrane damage caused by enterotoxigenic *Escherichia coli* (ETEC) by suppressing the delocalization of ZO-1 and dephosphorylating occludin. *Wang et al. (2016)* showed that *L. reuteri* LR1 prevented the disruption of ZO-1 caused by ETEC, thereby maintaining the barrier integrity of IPEC-1 cell. The role performed by occludin in TJs remains controversial, and one study reported that intestinal permeability in occludin-knockout mice was not altered, although the mice had unexplained clinical signs such as hyperplasia and chronic inflammation of the gastric epithelium (*Balda & Matter, 2016*). The OCEL domain, which mediates occludin interactions, is essential for limiting paracellular macromolecular flux (*Buschmann et al., 2013*). In the present study, dietary supplementation with *L. acidophilus* increased the mRNA expression and protein abundance of occludin, but did not affect ZO-1 mRNA expression and protein abundance in the jejunum and ileum. This indicates that *L. acidophilus* strengthens the TJs of the intestinal mucosa in weaned pigs, but other *Lactobacillus* strains may differ in their abilities to regulate TJs.

Intestinal barrier dysfunction is thought to be involved in the initiation of intestinal inflammation, and NLRs are thought to be involved in maintaining intestinal epithelial function and regulating intestinal inflammation (*Sardi et al., 2017*). NOD1 responses are potentially involved in both homeostatic and host defense responses to the presence of commensal and pathogenic organisms in the gastrointestinal tract. The immune response triggered by NOD1 is impaired at multiple levels in RIPK2-deficient mice (*Magalhaes et al., 2011*). *Majumdar, Nagpal & Paul (2017)* showed that W219R and L349P mutants had elevated mRNA expression of the pro-inflammatory cytokines IL-8 and IL-1

β after stimulation with the NOD1 ligand, as compared with the wild type NOD1, and these mutations may lead to increased NOD1 signaling and contribute to inflammation in ulcerative colitis. *Liu et al. (2017b)* reported that knocking-down NOD1 substantially inhibited the bacterially induced activation of NF-κB. There is extensive evidence that NOD1 responses to commensal organisms are not only important during the development of the mucosal immune system in early life (*Bouskra et al., 2008*), but also later on, in response to infection by important pathogens (*Watanabe et al., 2010*; *Asano et al., 2016*). NOD1 overexpression and RNA interference-induced suppression both demonstrate that NOD1 is critically involved in the chemokine secretion and NF-κB activation initiated by *L. monocytogenes* (*Opitz et al., 2006*). In the present study, *L. acidophilus* supplementation of the weaned piglets' diets reduced the mRNA expression of NOD1 signaling-related genes in the jejunum, but not in the ileum, which suggests that *L. acidophilus* activates different NLR signaling pathways in different tissues. The differential expression of NOD1 signaling-related gene expression in distinct intestinal tissues remains an incompletely understood process that requires further study.

A great deal of research has highlighted the vital function of NLRP3 in the intestinal barrier (*Mak'Anyengo et al., 2018*; *Pahwa et al., 2017*). Various endogenous and exogenous stimuli activate NLRP3 inflammatory corpuscles through different signaling pathways to activate caspase-1, which promotes the maturation and secretion of the inflammatory cytokines IL-1 β and IL-18 (*Miao et al., 2010*). *Bauer et al. (2010)* reported that NLRP3/ mice were markedly protected from dextran sodium sulfate (DSS)-induced colitis, and further studies found that the decline in IL-1 β secretion in these mice was dependent on lysosomal maturation and reactive oxygen species. However, it has also been reported that knocking-out caspase-1 led to the loss of epithelial integrity after DSS was administered to the mice (*Dupaul-Chicoine et al., 2010*). *Zaki et al. (2010)* reported massive leukocyte infiltration in the colons of NLRP3- or caspase-1-deficient mice. NLRP3-deficient mice were also more susceptible to infection with *Citrobacter rodentium* than wild-type mice (*Song-Zhao et al., 2014*), suggesting that NLRP3 expression has a protective effect on the intestinal epithelium. The observed discrepancies in these studies may be attributable to different experimental designs. An attractive hypothesis is that the time-dependent and early activation of NLRP3 induces pyroptosis during the early phase of inflammation (*Liu et al., 2013*). This hypothesis may explain the protective effect of NLRP3 inhibition on DSS-induced colitis. Recent research has shown that the regulatory network for NLR signaling is more complex than was initially thought. Our results show that *L. acidophilus* reduces the expression of IL-1 β and IL-18 pro-inflammatory cytokines in weaning piglets, perhaps by inhibiting the NLR signaling pathway. However, the exact mechanism by which *L. acidophilus* regulates NLRs in the intestines of weaned piglets awaits clarification.

## CONCLUSIONS

Supplementation with *L. acidophilus* improved the ADG of weaned piglets, reduced serum DAO activity, increased the mRNA expression and protein abundance of occludin in the jejunum and ileum tissues, inhibited NOD1-signaling-pathway-related genes expression

in the jejunum, and inhibited NLRP3-signaling-pathway-related genes expression in the jejunum and ileum tissues. Therefore, *L. acidophilus* may enhance the function of the intestinal barrier by down-regulating the expression of IL-1 β and IL-18 pro-inflammatory cytokines via the NOD1/NLRP3 signaling pathway in the weaned piglets from this study.

### Funding

This study was financially supported by the National Natural Science Foundation of China (Project NO. 31702147) and Tianjin Natural Science Foundation (Project NO. 18JCYBJC30000). The funders had no role in study design, data collection and analysis, decision to publish, or preparation of the manuscript.

### Grant Disclosures

The following grant information was disclosed by the authors:
National Natural Science Foundation of China: Project NO. 31702147.
Tianjin Natural Science Foundation: Project NO. 18JCYBJC30000.

### Competing Interests

The authors declare there are no competing interests.

### Author Contributions

- Shiqiong Wang conceived and designed the experiments, performed the experiments, analyzed the data, contributed reagents/materials/analysis tools, prepared figures and/or tables, authored or reviewed drafts of the paper, approved the final draft.
- Haihua Li contributed reagents/materials/analysis tools, authored or reviewed drafts of the paper.
- Chenhong Du analyzed the data.
- Qian Liu and Dongji Yang prepared figures and/or tables.
- Longbin Chen and Qi Zhu contributed reagents/materials/analysis tools.
- Zhixiang Wang conceived and designed the experiments, authored or reviewed drafts of the paper, approved the final draft.

### Ethics

The following information was supplied relating to ethical approvals (i.e., approving body and any reference numbers):

The animal care and use protocols were approved by the Review Committee for the Use of Institutional Animal Care and Use Committee of Henan Agricultural University (Protocol no. 20161030).

### Data Availability

The raw data are provided in the Supplemental Files.

## Supplemental Information

Supplemental information for this article can be found online at http://dx.doi.org/10.7717/peerj.6060#supplemental-information.

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
