# Peer review of "Effects of dietary supplementation with Lactobacillus acidophilus on the performance, intestinal physical barrier function, and the expression of NOD-like receptors in weaned piglets"

_PeerJ, doi:10.7717/peerj.6060_

## Round 0.1 · original submission · Major Revisions

Please address each of the comments made by the reviewers, or indicate your reason for not doing so where appropriate.

In particular, as suggested by Reviewer 2, please explain the number of technical and biological replicates used in the qPCR analysis. If more than one biological replicate was used, then please provide the full data set. If only one biological replicate was used, then please analyze the samples from the rest of the animals in the study.

As indicated by Reviewer 1, analysis of the TJ protein localization (e.g. using confocal microscopy) would strengthen the manuscript. The positioning of the TJ proteins is just as important as their abundance.

·

Basic reporting

This study mainly investigated the effect of dietary supplementation with Lactobacillus acidophilus on the performance, intestinal physical barrier function, and the expression of NOD-like receptors in weaned piglets. The results are positive.

Experimental design

the author should provide the reference for the bacterial concentration in this study.

Validity of the findings

Conclusion are well stated, linked to original research question & limited to supporting results.

Additional comments

1. the author should provide the reference for the bacterial concentration in this study.
2. in the table3, average daily weight gain was markedly enhanced, while no difference was noticed in final body weight. Please explain!
3. in fig1, the authors indicated that TGs were tested, while only zo1 and occludin were studed in this study, claudin family also should be incuded. In addition, mRNA in insufficient for investigating TGs as the protein abundances and the distribution in the intestine play a key role in the gut barrier function.
4. for the signalling study, western blot should conducted rather than PCR.

Reviewer 2 ·

Basic reporting

Some English language corrections are required. For example: (Line 198) “Weaning piglets will appear gastrointestinal physiological disorders….” The meaning is unclear and needs clarification
Intro and background show clear context, and is referenced with recent, relevant publications.
The structure conforms to PeerJ
Figures are relevant, high quality, and described. Labelling in the legends is confusing in respect to the replication. It was thought 8 animals were used in each group therefore n=8 and not n=3.
Raw data is supplied, but is incomplete. The raw data files relating to the qPCR results are either incomplete, or inaccurate. For example: in this study 16 animals were used with samples being taken from two regions (jejunum and ileum) of the intestine. For each tissue region, the expression of 10 genes was examined in triplicate by qPCR. Therefore, it would have been expected to find in the raw data files information appertaining to this. 10 genes, each measured in triplicate = 30 values. Multiply this by the 16 animals tested = 480 values per tissue resulting in a total of 960 values. However, all that is supplied in the raw data file is information relating to three technical replicates of the same sample and therefore n=1.

Experimental design

Original primary research with clearly defined research question. The knowledge gap has been identified and addressed. The study has sufficient technical replication (n=8 animals per group), and was conducted ethically.
Methods used require additional information to be replicated and also clarity is required. For example: (Line 125) “All piglets were weighed on day 14”. Unsure if this means day 14 of age, or post weaning, or something else. Please clarify.
For cDNA production and qPCR methodology the thermal profile used has not been described. It is also unclear whether SYBR or probes were used. Additionally, the reference gene used for the qPCR analysis has not been identified. It could be assumed that β-actin was used, but this needs to be clarified.

Validity of the findings

The results are confusing in the context of the number of replicates used. For example lines 116-119 states “sixteen crossbred healthy piglets were divided into 2 groups into eight replicate pens (n=8 each group)”. In line 160 for the qPCR it mentions “three replicates were performed for each reaction and results expressed as 2-ΔΔCT values”. This indicates technical replication of the same sample (n=1) and therefore intra-variation only. This only provides evidence on the ability to pipette accurately rather than any variation that may exist between individual animals within the groups. This also suggests that tissue from only a single animal from each group was used for downstream qPCR analysis and comparison. If this was indeed the case, how was it decided which animals were used for the analysis?
The conclusion is linked to the original research question, and is supported by the results provided. However, considering that the results are unclear it is not possible to determine how accurate this conclusion is.

Additional comments

Lactobacillus acidophilus is clearly beneficial in enhancing the growth performance of weaned piglets and reducing serum diamine oxidase activity. This work by Wang and colleagues provides evidence that an L. acidophilus strain isolated from pigs has a positive impact on the intestinal epithelial barrier by inducing the expression of specific TJ genes, and inhibiting the expression of genes of major signalling pathways. Overall the study is of a good design, with some interesting results obtained. I have some comments for further improvement of the manuscript.
Major points
1. In this study a total of 16 piglets were used which were divided into two groups, therefore 8 animals in each group. For the calculations of values for dietary treatments on growth performance (Table 3) and DAO and D-lactate (Table 4), you have clearly indicated that the values obtained were from all replicate animals in each group (n=8). Unfortunately, how the qPCR data was obtained and used in figures 1 and 2 is not clearly described. In line160 the authors state “three replicates were performed for each reaction and the results expressed as 2-ΔΔCT values”. This indicates technical replication of the same sample (n=1). This also suggests that tissue from only a single animal from each group was used for downstream qPCR analysis and comparison. If this was indeed the case, how was it decided which animals were used for the analysis? Could you please clarify what samples were used and what analysis was undertaken as it is currently unclear?
Minor points
1. Line 64: Authors described “The occludin protein blocks the cell gap”. It could be suggested to change this to “The occludin protein blocks the gap between cells”
2. Line 67: Remove “several” from “involved in several immune…”
3. Line 68: Remove “an” from “making it an important….”
4. Possibly include information relating to the importance of D-lactate and DAO in the introduction. As it is, there is no explanation as to why DAO was measured and the importance of D-lactate was not mentioned until the discussion
5. Line 108: The authors describe the “L. acidophilus strain used in this study as being isolated from the excrement of a healthy piglet”. However, how this bacterium was identified/characterised as L. acidophilus has not been described. If this information has previously been published include a reference, if not briefly explain how this strain was identified as such.
6. The thermal profile used for the qPCR needs to be described. Also, identify the reference gene used for the analysis.
7. Line 198: “Weaning piglets will appear gastrointestinal physiological disorders….” The meaning is unclear and needs clarification.
8. Line 213: Remove “for example” at the end of the sentence

---

## Round 0.2 · Minor Revisions

Thank you for the resubmission of your manuscript. I have reviewed your responses and believe the comments have been adequately addressed.

There is one comment in particular I would like to draw to your attention - that from Reviewer regarding the statistical analysis of the body weight gains. I can see from the data in Table 6 that the probiotic treatment group had a lower average initial weight and higher average final weight (neither significant), which meant that the probiotic group had a significantly higher weight gain. Please describe this in the results section to make it clearer to the reader.

Please also make the minor language correction indicated by the reviewer 2.

Reviewer 2 ·

Basic reporting

Some English language corrections are required. For example: (Line 221) “Weaned piglets will appear slow growth, a low feed utilization rate, and gastrointestinal physiological disorders such as diarrhea”. I suggest changing this to “Some weaned piglets have slow growth, a low feed utilisation rate …..”

Intro and background show clear context, and is referenced with recent, relevant publications.
The structure conforms to PeerJ
Figures are relevant, high quality, and described. Labelling in the legends has been clarified and relates back to the main text. Raw data is supplied.

Experimental design

Original primary research with clearly defined research question. The knowledge gap has been identified and addressed. The study has sufficient technical replication (n=8 animals per group), and was conducted ethically.
Methods used have been clarified and could be repeated.

Validity of the findings

The conclusion is linked to the original research question, and is supported by the results provided.

Additional comments

Lactobacillus acidophilus is clearly beneficial in enhancing the growth performance of weaned piglets and reducing serum diamine oxidase activity. This work by Wang and colleagues provides evidence that an L. acidophilus strain isolated from pigs has a positive impact on the intestinal epithelial barrier by inducing the expression of specific TJ genes, and inhibiting the expression of genes of major signalling pathways. Overall the study is of a good design, with some interesting results obtained.

---

## Round 0.3 · accepted · Accept

Thank you for making these minor changes. I am pleased to accept your manuscript for publication.

#